# GROUNDED HUMAN-ATTRIBUTED DESCRIPTION AND ACTIVITY RECOGNITION IN VIDEOS

## ABSTRACT

Understanding characteristics and activities of individuals in complex multi-person environments is crucial for real-world applications. However, existing datasets often simplify this problem to grounded group activity recognition, single-person activity recognition, or close-set activity recognition, thus limiting the generalizability of models trained on these datasets. In this work, we introduce the task of Grounded Human-Attributed Description and Activity Recognition (GHADAR), which involves describing the characteristics and activity descriptions of **every** person in a video, provided the location of the person in the video, which is more practical for real-world applications. To facilitate this, we introduce a new dataset derived from AVA-Actions by generating open-set captions for the person description and activity. In addition, we propose a novel method to effectively utilize the information contained in grounding during training by constraining the cross-attention masks during training in VLMs to improve performance for this task. Our experiments show that our method outperforms SOTA VLMs on this task. Finally, we demonstrate the limitations of existing evaluation metrics, which are overly reliant on human-annotations and exact text-text matching. As an added video-based evaluation, we propose a holistic VLM-based evaluation schema that compares concepts *directly* between the video and the generated predictions. Thus, in this work, we develop a complete framework for GHADAR, including a dataset, a novel method and an evaluation schema, thereby establishing a strong foundation for future research in this domain. Code and data will be made available post-review.

## 1 INTRODUCTION

The rapid progress of vision–language models (VLMs) has expanded the scope of computer vision from recognition to rich, multimodal understanding. Modern applications increasingly demand systems that can not only detect people in complex scenes, realized using powerful detectors like YOLO (Tian et al. (2025)) and DINO (Zhang et al. (2022)), but also describe their appearance, attribute fine-grained characteristics, and recognize ongoing activities in an open-set manner (Shin et al. (2024); Hossain et al. (2025); Chen et al. (2024a); Gu et al. (2018); Li et al. (2020)). Such human-attributed understanding is critical for intelligent home assistants, autonomous surveillance and checkout systems Sarkar & Kak (2021; 2024), and assistive technologies (Shin et al. (2024)).

We formalize this challenge as **Grounded Human-Attributed Description and Activity Recognition** (GHADAR): given the spatial location of people in a video (via bounding box), the goal is to generate two open-set captions — one describing *each person's attributes* (e.g., an elderly man in white on the street, a child in blue uniform) and one describing *their activity* (e.g., attempting to cross a road, playing on the footpath). This formulation demands *per-person* reasoning in multi-actor scenes, integrating both local appearance cues and broader contextual signals.

Traditionally, research efforts in this direction have typically reduced the complexity of human-centric video understanding due to the scarcity of annotated data as shown in Figure 1. For example, some works reduce this task to understanding the activity of a group of people as a whole or generating one caption for the entire video (Kim et al. (2023)). Meanwhile, some other works reduce the scope of activity recognition to a closed set of limited action labels (Li et al. (2020); Bian et al. (2025); Monfort et al. (2022)) or focus exclusively on single-person scenarios (Wang

Figure 1: Our proposed task with respect to existing tasks. While existing tasks are more relaxed, we propose an open-set human description and activity recognition that is more generic.

et al. (2025)). In response, this paper introduces a new dataset called AVA-Captions, which extends AVA-Actions (Gu et al. (2018)) with open-set, per-person captions for both attributes and activities, and involves multi-person interactions. Starting from AVA-Actions' per-person action labels, we employ a high-capacity VLM (Nova Pro v1 (AGI (2025))) to generate natural-language descriptions, followed by identity-aware deduplication via person re-identification. This enables training and evaluation of models on the proposed task.

We further propose **Constrained Attention Masking-based Pretraining (CAMP)**, a two-stage training strategy for VLMs that explicitly leverages grounding information. In Phase 1, we constrain the attention mask of the output tokens to the visual tokens within an expanded bounding box around the target person, enforcing localized reasoning. In Phase 2, we remove constraints and fine-tune at a reduced learning rate, allowing the model to integrate global context without eroding its person-specific focus.

Finally, we critique existing evaluation metrics in person and activity description. These metrics are largely text-text comparisons based on n-grams (BLEU, METEOR, ROUGE) (Papineni et al. (2002); Banerjee & Lavie (2005); Lin (2004)), or similarity between embeddings from pretrained models (SBERT, CLIP) (Radford et al. (2021); Reimers & Gurevych (2019)). Thus, these metrics often penalize semantically correct predictions that diverge lexically from human-annotated ground truth (Ohi et al. (2025)). To address this, we introduce a **VLM-driven evaluation framework** that directly compares conceptual alignment between the raw video content and the model prediction from the lens of completeness, uniqueness and correctness of the caption. In summary, our contributions are:

1. **Task and Dataset**: We define the task of Grounded Human-Attributed Description and Activity Recognition (GHADAR) and release AVA-Captions, the first large-scale dataset with open-set, per-person appearance and activity captions grounded in video
2. **Method**: We propose CAMP, a two-stage attention-masking pretraining strategy that explicitly leverages grounding for improved per-person reasoning.
3. **Evaluation**: We design a VLM-based video–prediction alignment protocol to complement classical text-matching metrics.
4. **Results**: We demonstrate consistent gains over strong VLM baselines on AVA-Captions and the publicly available HC-STVG v2 (Tang et al. (2022)) datasets, across both standard and proposed metrics.

## 2 RELATED WORK

**Existing Datasets:** While we are among the first to propose grounded appearance and activity captions for every person in a video, this task has been explored in various relaxed settings in the literature. We tabulate some related datasets and their characteristics in Table 1.HC-STVG (Tang et al. (2022)) contains multi-person video clips but annotates only a single target individual per clip. It provides natural-language descriptions of the subject's appearance and activity, along with frame-level bounding boxes, making it the closest existing benchmark to our setting. This dataset comes closest to our use case and hence, we also include results on this dataset in our experiments. VidSTG (Zhang et al. (2020); Shang et al. (2019)) is not a human-centric dataset and contains location captions for inanimate objects for most videos. In addition, it does not have person descriptions.

Table 1: Existing datasets. Multi-Activity represents whether captions are available for multiple activities done by the person in the video. Multi-Person represents whether annotations are available for multiple people appearing in the video. Most of the video clips in these datasets are short (0-20s) except for AVA-Actions and MEVA, which have 15 min long and 5 min long annotated videos, respectively. * - can be obtained with minimal processing

| Dataset | Size (# clips) | Human-Centric? | Grounded for every frame? | Open-Set Captions | Multi-Activity? | Multi-Person? |
|---|---|---|---|---|---|---|
| HC-STVG (Tang et al. (2022)) | 15K | ✓ | ✓ | ✓ | ✓ | |
| VidSTG (Zhang et al. (2020); Shang et al. (2019)) | 7.8K | | ✓ | ✓ | | |
| ActivityNet Entities (Zhou et al. (2019b)) | 15K | * | | ✓ | | ✓ |
| AVA-Actions (Gu et al. (2018)) | 450 | ✓ | ✓ | | ✓ | ✓ |
| AVA-Kinteics (Li et al. (2020)) | 230k | ✓ | | | ✓ | ✓ |
| MSR-VTT (Xu et al. (2016)) | 10K | * | | ✓ | | |
| RefAVA (Peng et al. (2024)) | 36K | ✓ | ✓ | | ✓ | ✓ |
| MEVA (Corona et al. (2021)) | 1.7K | ✓ | ✓ | | | ✓ |
| AVA-Captions (Ours) | 45K | ✓ | ✓ | ✓ | ✓ | ✓ |

ActivityNet Entities (Zhou et al. (2019b)) is not strictly human-centric but it is easy to filter out the other object annotations. However, it does not offer grounding information for all frames and a majority of the videos contain only one person in the field of view. AVA-Actions (Gu et al. (2018)) has frame-wise annotations per person. However, it does not have person descriptions and only provides action labels for 80 comprehensive, predetermined classes. We use this dataset to generate natural language descriptions for the person and their activity using these action labels. AVA-Kinetics (Li et al. (2020)) does not have open-set captions nor does it have bounding boxes for the persons in the video. RefAVA Peng et al. (2024) extends AVA-Actions by adding manually-annotated open set person descriptions. However, it does not provide open-set captions for human actions, which are required for understanding person-object interactions. Finally, MEVA (Corona et al. (2021)) is a human-and-group-centric datasets that has activity labels for individuals and group activities. However, this dataset offers a very limited range of activities and is more specific for the use-cases of surveillance. In summary, existing datasets either lack open-set captions, omit per-person grounding, or restrict to single-person or closed-set scenarios — motivating the need for AVA-Captions.

**Human Activity Recognition Methods:** Human activity recognition (HAR) is an essential component of fields like home assistants, embodied AI and assistive technology, and is gaining traction in Deep Learning (DL) literature (Shin et al. (2024); Hossain et al. (2025); Chen et al. (2024a)). Recent deep learning approaches treat HAR as a specialized form of video understanding, but often under constrained settings. (Wang et al. (2025); Kim et al. (2023); Wang et al. (2024); Kazakos et al. (2025)). For instance, (Wang et al. (2025)) combines local and global video processing to capture activity cues, but assumes only one person is present. (Kazakos et al. (2025)) performs large-scale pretraining for caption generation from videos. But this method is not necessarily human-centric and is more suited toward scene description rather than human action description. (Kim et al. (2023)) focuses on recognizing activities of people as a group rather than focus on individual actions, whereas (Bian et al. (2025)) focuses on HAR using the human pose. Some papers also approach this problem by predefining action classes and trying to classify human actions in one of these classes (Corona et al. (2021); Li et al. (2020); Bian et al. (2025); Monfort et al. (2022)). As opposed to these methods, which employ a relaxed setting for HAR, we are one of the first methods to propose the HAR task in a more unconstrained way, allowing open-set captions, and human-attributed action recognition that allows multiple people in the field of view.

## 3 AVA-CAPTIONS DATASET

Existing datasets focus on a single person or focus on a closed set of classes, as shown in Table 1. To address these limitations, we construct AVA-Captions, a large-scale benchmark for open-set, per-person appearance and activity description in video. We build AVA-Captions from AVA-Actions v2.2 (Gu et al. (2018)), which contains 15-minute annotated segments for 235 training and 64 validation videos. Each segment includes frame-level bounding boxes and action labels for every visible person, identified by a unique Person ID (PID), with actions drawn from 80 predefined classes. To create training samples, we segment each annotated video into 10-second clips with a maximum overlap of 5s, yielding approximately 36.8K training and 11.1K testing clips from the original training and validation splits. As AVA-Actions does not release test-set annotations, we do not use its official test split.

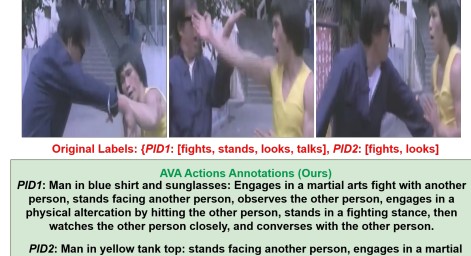

Figure 2: Examples from the AVA-Captions dataset. The captions in black are generated using expert-VLM while red action labels are provided in the original AVA-Actions dataset. More examples are provided in the appendix.

Once we have the extracted clips, we use a large VLM (called Expert-VLM) - Nova Pro (AGI (2025)), to generate separate captions for person and activity descriptions for each person in the clip. For the person description, we provide the video along with an image crop extracted using the bounding box around the person to the VLM and prompt it to generate a caption describing the person including their apparel, age and location (for eg. *a child in red hat standing near the stairs*). For the activity description, we also provide the activity labels provided in the original dataset to the expert VLM and ask it to construct a complete caption for the activity of the person using the video and the action labels. The exact prompts used for this process are provided in the appendix. This method can be adapted for other datasets as well to generate dense person-specific captions in the future.

However, there is one more concern regarding the AVA-Actions dataset. From our observations, we found that the same person is often assigned multiple PIDs. This occurs when the camera pans away from a person and comes back to the same person, who is then detected as a new entity. This causes the data to have multiple repeated entries for different PIDs since they represent the same person. To address this, we apply a **Person-ReID-based filter** using the torchreid library (Zhou & Xiang (2019); Zhou et al. (2019a; 2021)). If the image crop corresponding to a given PID has a ReID similarity score greater than $0.75$ with any previous PID, then we merge them and retain the previous PID. We use the Expert-VLM to merge the captions so that there is no repetition. Some examples are shown in Figure 2. We manually validated $1,000$ randomly sampled datapoints to assess the quality of captions generated by the VLM. For each video, we examined whether:

1. all ground-truth action labels were incorporated into the caption,
2. individuals were described accurately,
3. the annotated actions were correctly reflected, and
4. the same person was consistently assigned a single subject ID.

Our evaluation revealed that in approximately 95% of cases, the VLM successfully utilized all action labels and produced accurate person and activity descriptions. However, in roughly 15% of cases, the captions contained repeated actions, primarily due to duplicated action labels across timestamps in the original dataset. In addition, for around 15% of the cases, the same subject was still given multiple PIDs. Adjusting the ReID similarity score threshold to $0.85$ increased this number to 20%, whereas changing it to $0.65$ led to false merging of distinct entities into a single subject ID for around 30% cases. Thus, we choose $0.75$ as the threshold. While we note that VLM-based generation may introduce minor biases, our manual inspection suggests that such artifacts are limited and can be treated as noise without significantly impacting model development. Additional examples and dataset statistics are provided in the appendix.

## 4 CONSTRAINED ATTENTION MASKING-BASED PRETRAINING (CAMP)

### 4.1 PROBLEM FORMULATION

Given an input video and grounding information in the form of bounding boxes, the task of Grounded Human-Attributed Description and Activity Recognition expects a person description caption and

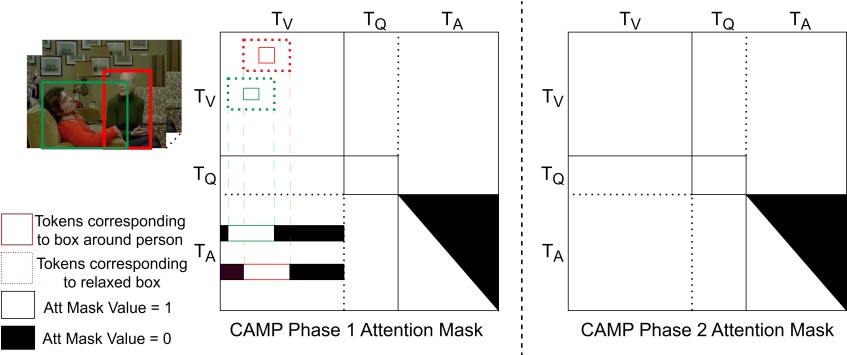

Figure 3: The two phases of CAMP. In the first phase, the modified causal mask for the answer tokens makes sure they can only attend to the vision tokens corresponding to the relaxed bounding box around the person for which the caption is being generated. In the second phase, the model is trained with a lower learning rate with a default attention mask, which allows insertion of global context into the attention weights. Here, $T_V$: vision tokens, $T_Q$: question tokens, $T_A$: Answer tokens. The illustration shows only one frame but the masking is done using all frames of the video.

an action caption for all people in the video. More formally, given video $V$, let there be $S$ number of subjects in the video. Let the sampled frames from the video be denoted by $\{F_1, F_2, ..., F_D\}$, where $D$ denotes the total number of frames. The bounding boxes around a subject $i$ are given by $B_i = \{b_1, b_2, ..., b_D\}$, where each element can be a bounding box in the $(x_1, y_1, x_2, y_2)$ format if the person is present in the frame or *None* otherwise. The bounding box format specifies the top-left and the bottom-right points of the bounding box in the frame. Thus, the complete grounding information can be denoted by $\mathbb{B} = \{B_1, B_2, ..., B_S\}$. Given this information, the model is expected to produce two outputs $C^P = \{c_1^P, c_2^P, ..., c_S^P\}$ and $C^A = \{c_1^A, c_2^A, ..., c_S^A\}$, where $C^P$ represents the person descriptions and $C^A$ denotes the activity captions of all the subjects in the video. Here, $c_i^P$ and $c_i^A$ denote the captions for person description and person activity for the subject.

In this study, we use a VLM for the model, which learns a function $f$ such that $A = f(V, Q | \mathbb{B})$, where $Q$ denotes the question prompt provided to the VLM and $A$ represents the output answer from the VLM. To model the above formulation, the question prompt includes information about the Person ID (PID) of the person and the bounding boxes. The output answer combines the person and activity descriptions in the following form: $A = $ "$\text{PID}_1. c_1^P : c_1^A; ...\text{PID}_S. c_S^P : c_S^A;$". The delimiters used in this format are useful for evaluation. Finally, an implicit constraint in this work is that the captions for person and activity description should be concise (a sentence instead of detailed descriptions), yet distinctive. This would help reduce further processing times for potential use cases. However, this is not a constraint that is inherent to the problem formulation and is enforced because of the ground truth annotations.

## 4.2 CAMP PHASE 1

In phase 1 of CAMP, we train the VLM InternVL3-1B (Chen et al. (2024b)) with a constraint on the attention mask during the decoding phase. In InternVL3, the video, question and answer are all converted into tokens using the VLM's image and text encoders. The attention mask, in turn, defines which token can attend to which token during the computation of attention weights in the transformer blocks of the VLM. In phase 1, we attempt to restrict the model to only look at regions around a person when generating output for the person.

More formally, let the number of video tokens be denoted by $T_V$, the number of question prompt tokens be denoted by $T_Q$, and the number of the ground truth answer prompt tokens be denoted by $T_A$. Then the attention mask for any layer of the VLM is a $N \times N$ matrix $Att$, where $N = T_V + T_Q + T_A$. In this matrix, the element $Att_{i,j}$ is given by:

$$Att_{i,j} = \begin{cases} 1, & \text{if token } i \text{ can attend to token } j \\ 0, & \text{otherwise} \end{cases} \quad (1)$$

By default, VLMs follow a causal attention masking strategy, where the output (answer) tokens can attend to all tokens before the current token, to prevent information leakage. Note that we are only interested in the answer tokens since the loss is computed only for the model outputs for these tokens. More formally, by default:

$$Att_{i,j} = \begin{cases} 0, & \text{if } i \leq T_V + T_Q \\ 1, & \text{if } j \leq i \\ 0, & \text{otherwise} \end{cases} \quad (2)$$

As mentioned previously, the annotated ground truth consists of captions from all people in the image. In the default setting, the output tokens for the parts corresponding to one person can also attend to the regions in the video not related to the person and the model is expected to learn the attention weights by itself to reduce the weightage of these regions. Instead, in phase 1 of CAMP, we explicitly constrain the model to generate outputs without attending to any regions except around the entity of interest during training. More formally, this translates to the following:

$$Att_{i,j} = \begin{cases} 0, & \text{if } i \leq T_V + T_Q \\ 1, & \text{if } j \leq i \text{ AND } j \in g_{ts}(i, \mathbb{B}) \\ 0, & \text{otherwise} \end{cases} \quad (3)$$

Here, $g_{ts}$ represents the token selection function that outputs a list of tokens that the current token can attend to. For video tokens, this function masks out the tokens that are unrelated to the person for which the output is being generated at token $i$. Thus, $g_{ts}(i, \mathbb{B}) = [k_1, k_2, ..., k_M]$, such that (i) $M \leq N$, (ii) $k_m \leq N \; \forall k_m \in g_{ts}(i, \mathbb{B})$, (iii) if $j > T_V, j \in g_{ts}(i, \mathbb{B})$, (iv) if $j \leq T_V, j \in g_{ts}(i, \mathbb{B})$ if and only if the token $j$ corresponds to the region in the video contained in the bounding boxes $B_p$ for person $p$ for which the caption is being generated at token $i$. While training, we fix the input size of each video frame to $448 \times 448$. Hence $g_{ts}(i, \mathbb{B})$ is a deterministic function. This is shown in Figure 3 where the attention mask values corresponding to the solid coloured lines are 1's.

However, this is not enough to capture the activity of the person since the bounding box around a person may not include the local context around the person, which is required to gauge the semantic activity of the person. Hence, we use a relaxed bounding box, as shown by the dotted coloured lines in Figure 3. The relaxed bounding box for a given bounding box $b$ is given by $[max(0, x_1 - w), max(0, y_1 - 0.2h), min(x_2 + w, 448), min(y_2 + 0.2h, 448)]$, where $w$ and $h$ represent the width and the height of the original bounding box $b$, and 448 denotes the size of the image. This function is motivated by the general observation in our dataset that the bounding boxes around people are generally higher than they are wider, and that significant context related to the activity is captured in the areas around the person.

## 4.3 CAMP PHASE 2

During Phase 1 training, we constrain the attention masks for each individual, encouraging the model to generate appearance and activity captions by focusing primarily on the local context surrounding the person. While effective for learning localized descriptions, this setup has two key limitations: (i) activities that inherently depend on global context—such as interactions with other people or distant objects—cannot be captured accurately under strict local constraints; and (ii) constructing person-specific attention masks requires knowing the exact mapping between output tokens and individuals, which is feasible during training but impractical at inference. A potential workaround would involve dynamically monitoring each generated token and updating the attention mask whenever a new subject is introduced, but this approach significantly increases inference complexity and latency, rendering Phase 1 alone unsuitable for deployment.

To address these issues, Phase 2 training removes the constraints and fine-tunes the model with the default attention mask (Figure 3) at a reduced learning rate. This continuation serves two purposes: it enables the model to incorporate global contextual cues while learning to generate outputs under the standard attention mechanism. At the same time, a lower learning rate ensures that that the attention weights of the model are not changed drastically so that the model still focuses on the regions around the person of interest. Together, these phases yield a model that balances localized grounding with global contextual reasoning, while remaining practical for inference.

Table 2: Comparison of CAMP against existing methods for Subject Description

| | AVA-Captions | | | | | HC-STVG | | | | |
|---|---|---|---|---|---|---|---|---|---|---|
| Method | BLEU4 (↑) | METEOR (↑) | ROUGE (↑) | CLIP Sim (↑) | SBert Sim (↑) | BLEU4 (↑) | METEOR (↑) | ROUGE (↑) | CLIP Sim (↑) | Sbert Sim (↑) |
| LLAVA-OV 0.5B | 28.29 | 54.22 | 59.47 | 0.75 | 0.54 | 18.70 | 53.94 | 57.47 | 0.86 | 0.48 |
| InternVL3-1B | 31.07 | 58.17 | 63.09 | 0.77 | 0.58 | 25.87 | 55.08 | 65.54 | 0.87 | 0.60 |
| Qwen2.5VL-3B | 36.46 | 63.17 | 68.06 | 0.82 | 0.65 | 25.66 | 55.07 | 65.81 | 0.88 | 0.62 |
| InternVL3 + CAMP (Ours) | 32.02 | 59.90 | 64.07 | 0.79 | 0.60 | 27.11 | **57.90** | 66.10 | **0.88** | 0.61 |
| Qwen2.5VL-3B + CAMP (Ours) | **36.48** | **63.62** | **68.43** | **0.83** | **0.67** | **27.33** | 57.84 | **67.82** | **0.88** | **0.64** |

## 4.4 MOTIVATION

We observed that when training a VLM on our dataset, the model frequently confused actions performed by different individuals (for example, action done by person B predicted for person A). To mitigate this, we introduce the CAMP framework, built on the hypothesis that for describing a person's actions and appearance, the local context surrounding the individual is generally sufficient, without requiring attention to the entire scene. By masking tokens outside the region of interest, the model is encouraged to learn mappings between localized visual features and their corresponding textual descriptions. While this hypothesis holds in many cases, modifying attention maps dynamically during inference is impractical. To address this limitation, we introduce Phase 2 training, where the constraints are removed and the model is fine-tuned with a reduced learning rate. Prior work has shown that lower learning rates yield more conservative weight updates and stabilize training (Lewkowycz (2021); Bjorck et al. (2021)). This second phase enables the model to incorporate global context while preserving its focus on the regions of interest, thereby eliminating the need for attention map modifications at inference time.

## 5 EXPERIMENTS

**Comparison with Similar Scale VLMs:** We perform experiments on the HC-STVG and AVA-Captions datasets with baseline, open-source VLMs and our CAMP method. The experimental setup is detailed in the appendix. We tabulate the results of CAMP with InternVL3-1B Chen et al. (2024b) and Qwen2.5VL-3B Bai et al. (2025) backbone as compared to similar open source baselines in Tables 3 and 2. We see a clear improvement in all classical and embedding-based metrics for both datasets. Table 2 evaluates the model prediction for subject description while Table 3 evaluates the model prediction for activity description. As described before, the output is of the format: $A =$ "$\text{PID}_1. c_1^P : c_1^A; ...\text{PID}_S. c_S^P : c_S^A; $". We separate this output into subject description and activity description using the colon delimiter. For different subjects, we use the PID to map the predictions to the ground truth. The computed metrics are averaged across all videos and subjects. As compared to the LLAVA-One Vision (LLAVA-OV) baseline (Li et al. (2024); Liu et al. (2023)), which offers a 0.5 B model, we see a significant improvement with InternVL3 1B model. While some of this can be attributed to the more number of parameters (since LLAVA-OV does not offer 1B model), the multimodal encoder structure in InternVL3 (Chen et al. (2024b)) style of models also plays a significant part. Unlike LLAVA, both InternVL3 and Qwen support frame-level injection, enabling bounding boxes to be passed separately for each frame. This design yields consistent improvements, which are further amplified by our CAMP framework through constrained attention in Phase 1 and unconstrained fine-tuning in Phase 2. Interestingly, we observe greater relative improvements for the smaller InternVL3-1B backbone compared to the larger Qwen2.5VL-3B. This is expected: larger models can already capture complex functions, making constrained training less impactful, whereas smaller models benefit more from CAMP's inductive bias. Due to compute limitations, we were unable to compare with larger-scale models. However, we emphasize that the proposed task is most relevant for home assistants and surveillance systems, where inference typically occurs on edge devices. For this reason, our study focuses on small- to medium-scale models, which are more practical for such applications.

**Ablation on Relaxation of Bounding Box:** During the attention masking in Phase 1 of CAMP, we use a relaxed bounding box around the person to capture local context for predicting activity. We expand it by 100% of the width on each side and by 20% of the height on each side based on the observation that humans are generally taller than wider. Hence, the bounding box around them captures lesser context along the width than the height. Thus, for the relaxed bounding box, we

Table 3: Comparison of CAMP against existing methods for Activity Description

| Method | AVA-Captions | | | | | HC-STVG | | | | |
|---|---|---|---|---|---|---|---|---|---|---|
| | BLEU4 (↑) | METEOR (↑) | ROUGE (↑) | CLIP Sim (↑) | SBert Sim (↑) | BLEU4 (↑) | METEOR (↑) | ROUGE (↑) | CLIP Sim (↑) | Sbert Sim (↑) |
| LLAVA-OV 0.5B | 13.39 | 37.71 | 39.69 | 0.85 | 0.53 | 8.34 | 32.43 | 41.78 | 0.82 | 0.40 |
| InternVL3-1B | 19.34 | 44.91 | 46.28 | 0.89 | 0.63 | 8.60 | 32.66 | 38.03 | 0.85 | 0.42 |
| Qwen2.5VL-3B | 19.30 | 43.02 | 47.35 | 0.90 | 0.63 | 8.97 | 31.52 | 40.01 | 0.86 | 0.47 |
| InternVL3 + CAMP (Ours) | 20.11 | **46.15** | 46.82 | 0.90 | **0.65** | **11.17** | **35.67** | **42.23** | **0.87** | 0.46 |
| Qwen2.5VL-3B + CAMP (Ours) | **20.14** | 43.40 | **47.90** | **0.91** | **0.65** | 10.64 | 33.08 | 40.65 | 0.86 | **0.47** |

Table 4: Effect of relaxations of bounding box during CAMP on HC-STVG dataset. The columns represent the one-sided changes to width ($\Delta w$) and height ($\Delta h$) in terms of percentage of the total width and height. For example, (20,20) signifies 20% relaxation in width and height on each side, thus making the relaxed bounding box as $[max(0, x_1 - 0.2w), max(0, y_1 - 0.2h), min(x_2 + 0.2w, 448), min(y_2 + 0.2h, 448)]$

| | ($\Delta w, \Delta h$) | | | | | | | |
|---|---|---|---|---|---|---|---|---|
| | (0,0) | (20,20) | (50,50) | (100,100) | (100,0) | (100,20) (used) | (100,50) | (20,50) |
| Subject Sbert | 0.54 | 0.54 | 0.57 | 0.58 | 0.60 | **0.61** | 0.60 | 0.55 |
| Subject METEOR | 52.21 | 52.54 | 54.96 | 57.09 | 56.98 | **57.90** | 56.59 | 53.64 |
| Subject ROUGE | 61.73 | 61.67 | 63.88 | 65.87 | **66.38** | 66.10 | 65.92 | 62.30 |
| | | | | | | | | |
| Activity Sbert | 0.44 | 0.44 | 0.45 | 0.46 | 0.46 | **0.46** | 0.46 | 0.44 |
| Activity METEOR | 33.32 | 33.04 | 34.00 | 34.40 | 35.00 | **35.67** | 34.87 | 33.55 |
| Activity ROUGE | 40.53 | 40.44 | 41.40 | 41.84 | 42.06 | **42.23** | 41.87 | 41.07 |

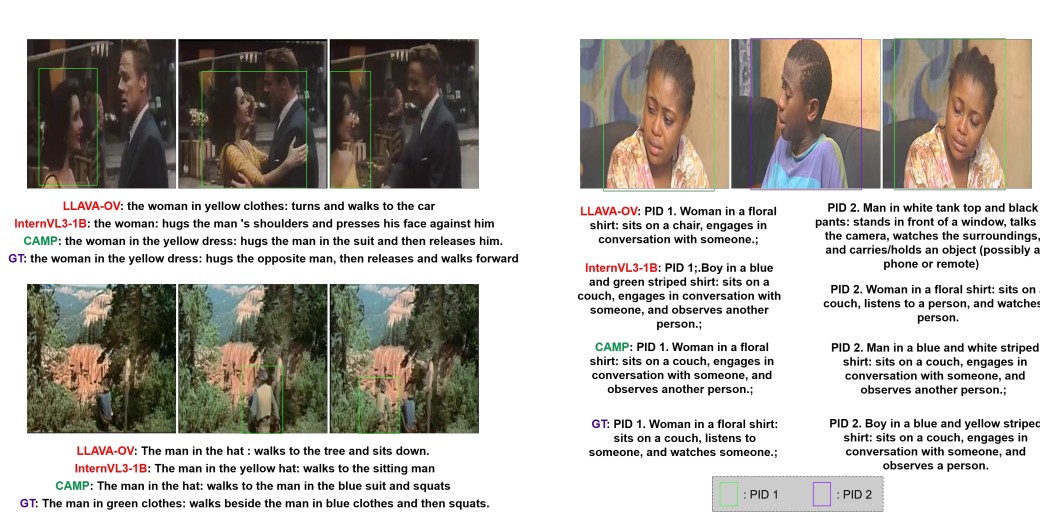

Figure 4: Qualitative examples from HC-STVG (left two) and AVA-Captions (right). HC-STVG has captions only for one person, whereas AVA-Captions has captions for both people in the video.

expand it more along the width. For added validation, we vary the proportions of these relaxations in Table 4. We observe that increasing the context in width is more effective than height as expected.

**Ablation on the phases of CAMP** is provided in the appendix due to space constraints. We find that employing only Phase 1 causes 10% drop due to lack of global context, while directly going to Phase 2 without the first phase causes a drop of 4%. This shows the necessity of both phases.

# 6 VLM-BASED METRICS

**Metric Definition:** Text-text matching or embedding similarity metrics are inherently tied to the specific ground truth (GT) annotations (Ohi et al. (2025)). As a result, they can undervalue predictions that are semantically correct but differ lexically from the GT, or that capture details omitted in the annotation. For example, consider Figure 5 which contains low-scoring predictions from our model. However, it can be seen from the video frame that the events described by the prediction are

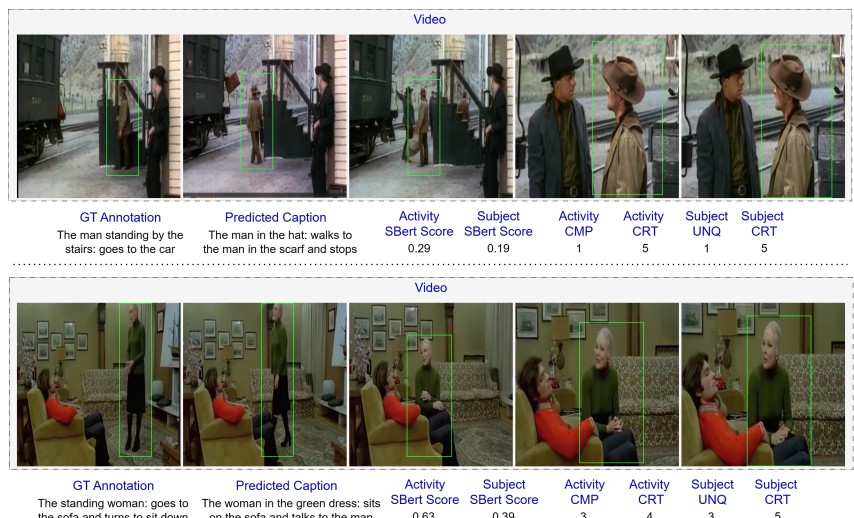

Figure 5: Examples of predictions wrongly given low metrics. In the first row, the predicted events occur in the video but are not captured by the GT annotation, leading to low SBert scores. Similarly, in the second row, the GT and predicted ways to describe the woman are valid, yet subject SBert score is low due to lexical mismatch. This necessitates the need for VLM-based metrics. More examples in the appendix

actually happening and noteworthy. We provide more such examples in the appendix. This highlights a limitation: existing metrics cannot directly verify whether a predicted caption is factually grounded in the visual content. To address this, we propose using a large-scale model to act as a judge, given the actual video and the predictions. Thus, these metrics are computed without knowing the GT captions and designed to complement, rather than replace, the classical measures. More specifically, given a video, an image crop of the target subject, and the predicted caption, we query Nova-Pro v1 (AGI (2025)) as well as Qwen2.5 VL 32B (Bai et al. (2025)) to assign a score from 1–5 (higher is better) for each of the following:

1. Subject / Activity Correctness (CRT): How much of the predicted caption is actually present in the video for the given subject. CRT measures factual alignment of the prediction with the video content.
2. Subject / Activity Completeness (CMP): How much of the subject description / activity is described by the caption. CMP measures whether all relevant actions are captured by the prediction.
3. Subject Uniqueness (UNQ): How discriminative is the predicted subject description in identifying the subject. UNQ measures ambiguity in the predicted caption.

CRT and CMP are analogous to precision and recall, respectively whereas UNQ is motivated for practical use cases and its definition is dependent on the video. For example, if a predicted caption reads: "The man : opens the door", it can be considered highly unique for describing the subject if there is only one man in the video, but not at all unique when there are multiple men in the video. We believe that UNQ is a better metric than CMP for subject description for practical purposes. In addition, CMP is ill-defined especially for the subject description since there can be various ways of describing the same person. While this is also true for the activities, CMP gives an approximate measure of whether all important activities are captured by the model and so we retain it as a metric. Some examples are shown in Figure 5. Based on these definitions, we compute (i) the average CRT, (ii) average CMP, (iii) average UNQ and (iv) accuracy based on $CRT \geq 4$ AND $UNQ \geq 3$ for subject and $CRT \geq 4$ AND $CMP \geq 3$ for activity (Acc), as VLM-based metrics. The full prompt used for querying the VLM for computing these metrics is provided in the appendix.

**Comparison with Similar Scale VLMs:** We tabulate the subject description metrics in Table 5 using Nova Pro v1 and Qwen2.5 VL 32B as expert models respectively. Table 6 show activity caption metrics using Nova Pro v1 and Qwen2.5 VL 32B as expert models respectively. We see that our method consistently outperforms baselines and is able to produce better and more comprehensive

Table 5: Comparison of CAMP against existing methods for Subject Description with VLM-based metrics using Nova Pro v1 and Qwen2.5 VL 32B as expert VLMs

| VLM Evaluator | Nova Pro v1 | | | | | | Qwen2.5 VL 32B | | | | | |
|---|---|---|---|---|---|---|---|---|---|---|---|---|
| Dataset | AVA-Captions | | | HC-STVG | | | AVA-Captions | | | HC-STVG | | |
| Method | Avg CRT (↑) | Avg UNQ(↑) | Acc (↑) | Avg CRT (↑) | Avg UNQ (↑) | Acc (↑) | Avg CRT (↑) | Avg UNQ(↑) | Acc (↑) | Avg CRT (↑) | Avg UNQ(↑) | Acc (↑) |
| LLAVA-OV 0.5B | 2.84 | 2.06 | 0.29 | 3.99 | 1.94 | 0.34 | 2.89 | 2.29 | 0.38 | 2.86 | 2.46 | 0.30 |
| InternVL3-1B | 3.03 | 2.07 | 0.32 | 4.03 | 2.09 | 0.40 | 3.05 | 2.34 | 0.41 | 3.03 | 2.47 | 0.37 |
| Qwen2.5VL-3B | 3.46 | 2.35 | 0.43 | **4.36** | 2.04 | 0.39 | 3.44 | 2.58 | 0.66 | 3.25 | 2.53 | 0.33 |
| InternVL3 + CAMP (Ours) | 3.09 | 2.11 | 0.34 | 4.29 | **2.25** | **0.48** | 3.13 | 2.38 | 0.43 | **3.28** | **2.69** | **0.45** |
| Qwen2.5VL-3B + CAMP (Ours) | **3.51** | **2.36** | **0.43** | 4.27 | 2.11 | 0.41 | **3.47** | **2.63** | **0.67** | 3.29 | 2.56 | 0.38 |

Table 6: Comparison of CAMP against existing methods for Activity Description with VLM-based metrics using Nova Pro v1 and Qwen2.5 VL 32B as expert VLMs

| VLM Evaluator | Nova Pro v1 | | | | | | Qwen2.5 VL 32B | | | | | |
|---|---|---|---|---|---|---|---|---|---|---|---|---|
| Dataset | AVA-Captions | | | HC-STVG | | | AVA-Captions | | | HC-STVG | | |
| Method | Avg CRT (↑) | Avg CMP(↑) | Acc (↑) | Avg CRT (↑) | Avg CMP (↑) | Acc (↑) | Avg CRT (↑) | Avg CMP(↑) | Acc (↑) | Avg CRT (↑) | Avg CMP(↑) | Acc (↑) |
| LLAVA-OV 0.5B | 3.24 | 2.39 | 0.54 | 2.21 | 1.17 | 0.08 | 3.59 | 3.17 | 0.75 | 3.07 | 2.43 | 0.53 |
| InternVL3-1B | 3.54 | 2.65 | 0.64 | 3.17 | 1.94 | 0.41 | 3.80 | 3.40 | 0.83 | 3.93 | 3.28 | 0.88 |
| Qwen2.5VL-3B | 3.61 | 2.83 | 0.68 | **3.51** | 2.07 | 0.48 | 3.86 | 3.51 | 0.85 | 3.96 | 3.18 | 0.87 |
| InternVL3 + CAMP (Ours) | **3.63** | 2.75 | 0.68 | 3.45 | **2.16** | **0.50** | **3.92** | 3.51 | **0.86** | **4.04** | **3.31** | **0.90** |
| Qwen2.5VL-3B + CAMP (Ours) | **3.63** | **2.86** | **0.71** | 3.43 | 2.05 | 0.46 | 3.86 | **3.52** | 0.85 | 4.02 | 3.20 | 0.89 |

outputs as shown in Figure 4. Even though the average CRT, UNQ, or CMP rise slightly, it generates a significant difference in accuracy after thresholding, which is due to the discrete nature of the VLM-based metrics. As seen in the examples, constrained attention using CAMP can guide the VLM towards localizing actions for a given individual instead of getting confused with the global context alone. For example, without CAMP, InternVL3 confuses between the two PIDs, whereas LLAVA-OV confuses the person description.

**Limitations:** Recent works have shown the effectiveness of using VLMs as judges Gu et al. (2025); Lee et al. (2024) where the proximity of the VLM results to the human results is heavily dependent on the prompt. However, VLMs can introduce inherent bias in their results as described in detail in the Appendix. Hence, we recommend using VLM-based metrics for relative comparison between models rather than the absolute performance of a single model.

**Role of LLMs:** We used LLMs for data annotation and evaluation as a part of our approach. We have cited the LLMs and provided the implementation details including the exact prompt used for the task in the paper and appendix.

## 7 CONCLUSION

In this paper, we introduce the task of Grounded Human-Attributed Description and Activity Recognition and propose a dataset called Ava-Captions for the same by modifying the AVA-Actions dataset. In order to effectively use the grounding information in this task, we propose the CAMP method that constrains the attention mask during training, thereby directing the model to focus more on the local context around a person to better caption their description and activity. Finally, we propose VLM-based metrics that directly compares predicted captions with the content of the video instead of incomplete human annotations and serves as a complement to classical metrics. A limitation in the VLM-based metric is that it is highly dependent on the ability of the VQA to understand the video and the instruction. With this work, we want to encourage future research in the human-attributed video understanding tasks that can be useful for various applications.

### ETHICS STATEMENT

AVA-Captions is an extension of the original AVA-Actions dataset. We use the same videos as the original and modify the annotation files. It is required to cite both datasets (ours and the original AVA-Actions) if using our dataset.

REPRODUCIBILITY STATEMENT

We will be releasing the code and the AVA-Captions dataset publicly after the review. We have provided the exact prompts used during the generation process in the appendix Section A. We have provided the running environment and setup in Appendix Section B. For evaluating the method, we use standard metrics and VLM-based metrics. For the latter, we use and present results with two different VLMs. We use the same prompts for both cases, which are provided in Appendix section D. We will also be providing the model checkpoints and prediction files when we release the dataset and code to ensure reproducibility.

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

## A  PROMPT FOR GENERATING AVA-CAPTIONS

We use Nova Pro v1 as an expert model to generate full captions about person and activity descriptions, given the video, cropped image of the person and the activity labels provided in the AVA-Actions dataset. For multiple people in the video, the expert model is run for each person and the resulting captions are merged. The exact prompt used for this purpose is as follows:

*<VIDEO> <PERSON-CROP> This is a 10 seconds video and an image. Consider the person in the image. The action labels associated with the person in the video are given by "<ACTION1>,<ACTION2>,...". These actions describe only one person. Describe the selected person in regards to clothing, location or other surprising appearance, for example "Sitting blindfolded woman in a black suit". Use all the action labels to create an activity caption only for the selected person in the video while maintaining the temporal action order. The output should be in the format: <person description>: <activity caption>. For example, "Blindfolded woman in black suit: sits on a chair, gets up, opens the door, hugs the man at the door and walks out".Structure the action caption in a conceptual manner, with slightly more focus on the meaning of actions rather than simple words.*

Table 7: Statistics for the number of people per video clip in the AVA-Captions dataset

| Phase | Total video clips | Minimum # People | Maximum # People | Mean # People | Median # People |
|---|---|---|---|---|---|
| Training | 36815 | 1 | 10 | 1.96 | 2 |
| Testing | 11095 | 1 | 11 | 1.98 | 2 |

Table 8: Ablation on the phases of CAMP. The first row denotes the baseline without CAMP, next row denotes only Phase 1 without Phase 2 and last row denotes the best method, which includes both Phase 1 and Phase 2,

| | | HC-STVG Subject | | | | | HC-STVG Activity | | | | |
|---|---|---|---|---|---|---|---|---|---|---|---|
| Phase 1 | Phase 2 | BLEU4 (↑) | METEOR (↑) | ROUGE (↑) | CLIP Sim (↑) | SBert Sim (↑) | BLEU4 (↑) | METEOR (↑) | ROUGE (↑) | CLIP Sim (↑) | SBert Sim (↑) |
| ✓ | | 20.09 | 45.05 | 52.65 | 0.58 | 0.36 | 5.15 | 23.88 | 24.55 | 0.57 | 0.26 |
| | ✓ | 25.87 | 55.08 | 65.54 | 0.87 | 0.60 | 8.60 | 32.66 | 38.03 | 0.85 | 0.42 |
| ✓ | ✓ | **27.11** | **57.90** | **66.10** | **0.88** | **0.61** | **11.17** | **35.67** | **42.23** | **0.87** | **0.46** |

| | | AVA-Captions Subject | | | | | AVA-Captions Activity | | | | |
|---|---|---|---|---|---|---|---|---|---|---|---|
| Phase 1 | Phase 2 | BLEU4 (↑) | METEOR (↑) | ROUGE (↑) | CLIP Sim (↑) | SBert Sim (↑) | BLEU4 (↑) | METEOR (↑) | ROUGE (↑) | CLIP Sim (↑) | SBert Sim (↑) |
| ✓ | | 10.27 | 20.81 | 29.74 | 0.59 | 0.49 | 6.93 | 20.81 | 18.49 | 0.64 | 0.46 |
| | ✓ | 31.07 | 58.17 | 63.09 | 0.77 | 0.58 | 19.34 | 44.91 | 46.28 | 0.89 | 0.63 |
| ✓ | ✓ | **32.02** | **59.90** | **64.07** | **0.79** | **0.60** | **20.11** | **46.15** | **46.82** | **0.90** | **0.65** |

## B  EXPERIMENTAL SETUP

We perform experiments on the HC-STVG and AVA-Captions datasets with baseline, open-source VLMs and our CAMP method. We use 3 A100-SXM4s for our experiments with a batch size of 3. We use an image size of 448X448 and sample 10 equidistant frames from the video. Since most videos are 10 seconds long, this essentially corresponds to an effective sampling rate of 1 FPS. For evaluation, we calculate the subject and activity metrics separately. Subject metrics evaluate the predicted human appearance captions and activity metrics evaluate the predicted activity captions against the ground truth. We use the standard metrics for text-text comparison, namely BLEU4 (Papineni et al. (2002)), ROUGE (Lin (2004)) and METEOR (Banerjee & Lavie (2005)). In addition, we also calculate the embedding similarity metrics namely CLIP Similarity Score (CSS) and SBert Similarity Score (SbSS). For these, we compute the CLIP (Radford et al. (2021)) embeddings and SentenceBert (Reimers & Gurevych (2019)) embeddings, respectively, for the predicted caption and the ground truth caption, and compute the cosine similarity between the two.

## C  MORE EXAMPLES FROM AVA-CAPTIONS DATASET

Figure 6 shows examples from the AVA-Captions dataset. The VLM is able to accurately describe the person description even though it is not provided in the original ground truth. At the same time, the activity caption follows the action labels provided in the ground truth but is an open-set caption. Data statistics are provided in Table 7.

## D  ABLATION ON PHASES OF CAMP

We perform an ablation with just Phase 1 of CAMP, where we apply constrained attention masking during training as well as inference on HC-STVG and AVA-Captions dataset. The results are shown in Table 8. During inference, we keep a conditional statement at every step of the generation that checks for special tokens for change of subject and modifies the attention mask accordingly. However, we find significantly lower scores on both datasets. This is not surprising since false outputs can lead to false attention masks for all the tokens after the false output.

## E  MOTIVATING THE NEED FOR VLM-METRICS

We present more examples of cases where the ground truth for a given subject in a video lexically differs from the equally correct predictions in Figures 7 and 8. A video contains a lot of information,

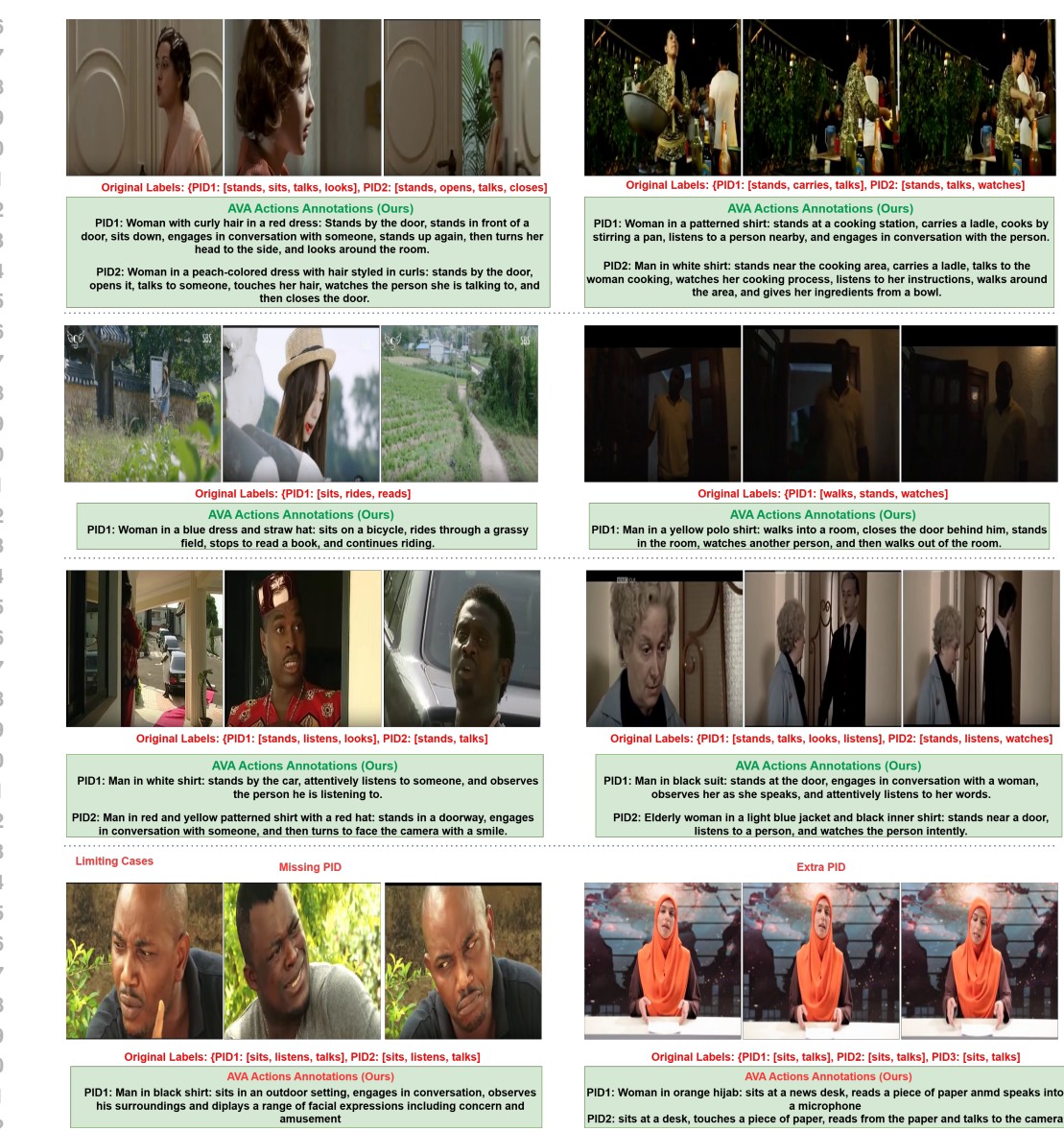

Figure 6: Examples from the AVA-Captions dataset. The captions in black are generated using expert-VLM while red action labels are provided in the original AVA-Actions dataset.

not all of which is captured by a human annotator in most cases. Thus, classical metrics which directly compare with these are significantly pessimistic. This motivates the need for VLM-based metrics, where the video is directly analyzed to assess the predicted caption.

# F    LIMITATIONS OF VLM-BASED EVALUATION

While foundation models today perform extremely well for visual understanding tasks, using them for evaluation can still induce biases in the result. Consider the example shown in Figure 8. Here, the caption describes the subject as "the standing man". Since no other men are standing in the video, it should get a high UNQ value, which is not the case. Similarly, some of the activities of the person are accurately described, yet the VLM gives the lowest CMP scores to the prediction. In our observation, especially for CMP and UNQ, the VLM is biased towards more pessimistic values for these metrics.

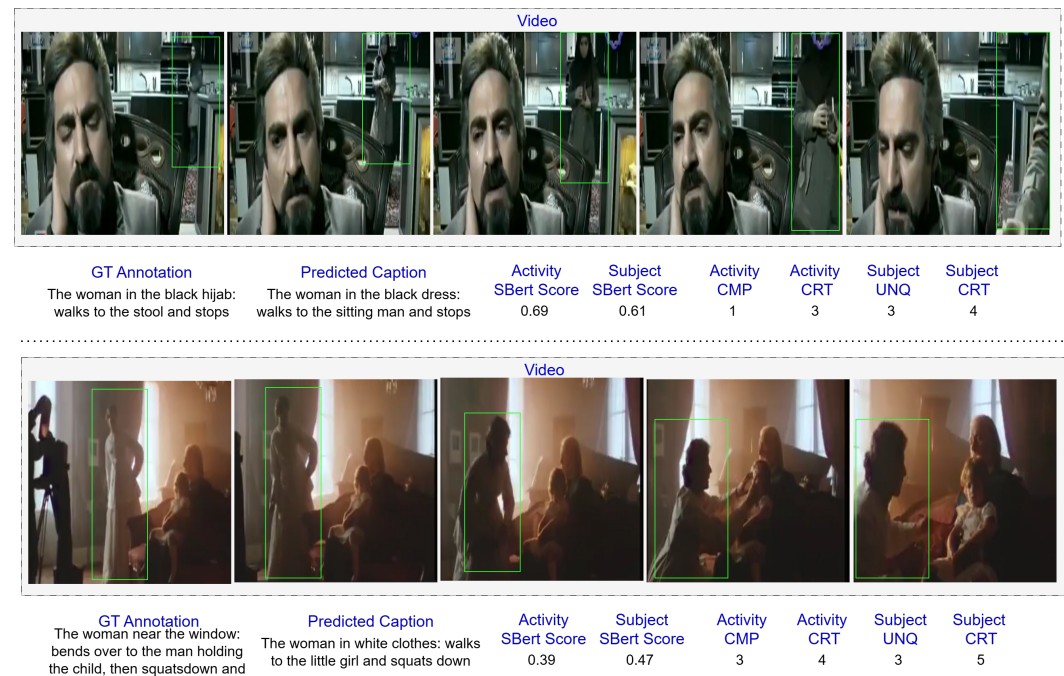

Figure 7: Examples of predictions which are lexically dissimilar from the ground truth but are equally valid for the given video

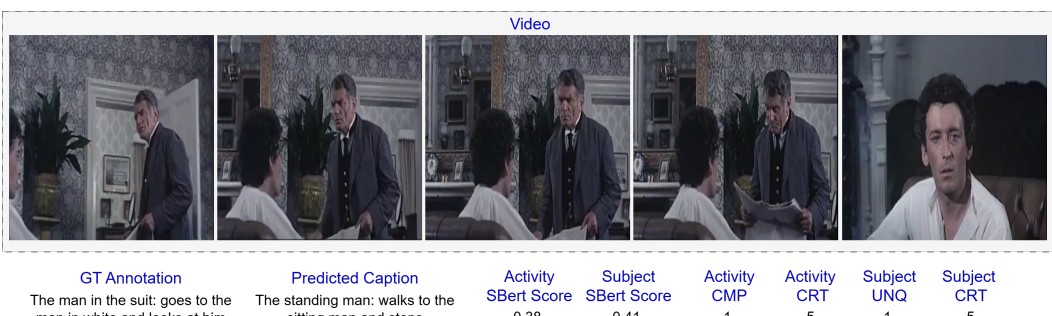

Figure 8: A limiting case of VLM-based evaluation metrics. CMP and UNQ are pessimistically given a score of 1 even though the predicted caption describes the person uniquely and describes some of the activities of the subject.

Thus, while VLMs offer crucial insights complementing the classical metrics, it is best to use the VLM-metrics for relative comparison between models rather than absolute performance of a model.

## G PROMPT FOR VLM-BASED METRICS

We used the Nova Pro v1 model to generate the CRT, CMP, and UNQ metrics. Since UNQ is not suitable for activity metrics, we run the VLM separately for the subject and the activity. Since HC-STVG dataset has annotations for only a certain portion of the video, we also provide the starting and ending timestamp in the prompt. The exact prompts are as follows:

**Subject:** *Consider only t=`<start time>` to t=`<end time>` seconds in the video. Consider only the appearance related details of the Person ID 1: `<subject description from ground truth>`. Next, consider this candidate caption which we want to evaluate against your caption and the events that are discernible from the video from t=`<start time>` to t=`<end time>` seconds:"`<predicted subject>`"*

*Your task is to rate the candidate caption for Person ID 1 in the video on a scale of 1 to 5 for each of the following criteria and rating scale INDEPENDENTLY.*

*Evaluation Criteria:- Completeness: How completely does the caption cover all the relevant and essential aspects of Person ID 1 in the video?*

*Rating Scale:*

*- 1 Very Low Completeness: The caption includes almost none of the important attributes for the person, missing most details.*

*- 5 Extremely High Completeness: The caption fully covers all relevant and essential aspects, leaving no important details unaddressed.*

*Evaluation Criterion: - Correctness: How accurately does the caption describe the aspects and activities of Person ID 1?*

*Rating Scale: - 1 Very Low Correctness: The caption is mostly or entirely incorrect; The person cannot be described by any or most of the predicted attributes*

*- 5 Extremely High Correctness: All attributes mentioned in the predicted caption belong to the person in the video.*

*Note that a caption may be correct but not complete and vice-versa. For example, if a caption has 2 attributes while the person requires 6 attributes to be uniquely defined, it is low complete. But if those 2 predicted attributes actually belong to the person, it has extremely high correctness. Similarly, if the predicted caption has 3 attributes and the person has only one of them and nothing else, the caption has high completeness since the actual attribute was captured, but low correctness because incorrect attributes were also predicted. Thus, completeness is similar to recall and correctness is similar to precision.*

*Evaluation Criterion: - Uniqueness: First, look at the different people in the video. Given the candidate caption, how easy it is to locate Person ID 1 in the video? Note that a simple caption can be highly discriminative. For example, if the video has only men and one woman, a caption like "a woman" is extremely unique as it can only be used to describe that one woman in the video.*

*Rating Scale:*

*- 1 Very Low Uniqueness: The same caption can be used to describe multiple entities in the video*

*- 5 Extremely Uniqueness: There cannot be any shred of doubt about which person from the video is being described by the caption.'*

**Activity:** *Consider only t=<start time> to t=<end time> seconds in the video. Consider only the activities of the Person ID 1: <subject description from ground truth>. Next, consider this candidate caption which we want to evaluate against your caption and the events that are discernible from the video from t=<start time> to t=<end time> seconds: "<predicted activity caption>"*

*Your task is to rate the candidate caption for Person ID 1 in the video for each of the following criteria and rating scales INDEPENDENTLY.*

*Evaluation Criteria:- Completeness: How completely does the caption cover all the relevant activities of Person ID 1 in the video?*

*Rating Scale:*

*- 1 Very Low Completeness: The caption includes almost none of the important activities, missing most activity details.*

*- 5 Extremely High Completeness: The caption fully covers all relevant and essential aspects, leaving no important details unaddressed.*

*Evaluation Criterion: - Correctness: How accurate is the caption regarding the activities of Person ID 1?*

*Rating Scale:*

*- 1 Very Low Correctness: The caption is mostly or entirely incorrect; The person is not doing any or most of the predicted activities*

*- 5 Extremely High Correctness: Almost all activities mentioned in the predicted caption are done by the person in the video.*

*Note that a caption may be correct but not complete and vice-versa. For example, if a caption has 2 activities while the person is doing 6 activities, it is low complete. But if those 2 predicted activities are actually done by the person, it has extremely high correctness. Similarly, if the predicted caption has 3 activities and the person is doing only one of them, the caption has high completeness since the actual activity was captured, but low correctness because incorrect activities were also predicted. Thus, completeness is similar to recall and correctness is similar to precision.*

