# OpenReview forum: "Grounded Human-Attributed Description and Activity Recognition in Videos"
_ICLR.cc/2026/Conference — Submitted to ICLR 2026_

### Official Review · Reviewer_HH6R · 2025-10-27

**Soundness:** 2
**Presentation:** 2
**Contribution:** 2
**Rating:** 4
**Confidence:** 4

**Summary:**

This work introduces the task of Grounded Human-Attributed Description and Activity Recognition (GHADAR) in videos, which aims to generate captions containing human appearance and activity given their locations. To facilitate this task, the authors construct the AVA-Captions dataset based on the AVA-Actions using a VLM and a person re-identification mechanism. The paper further proposes Constrained Attention Masking-based Pretraining (CAMP), an approach to improve VLMs for GHADAR, and provides an evaluation schema for GHADAR. Experiments on AVA-Captions and HC-STVG demonstrate that CAMP outperforms selected baselines in this task.

**Strengths:**

1. The completeness and workload of this work are commendable. The authors present a full pipeline covering problem definition, dataset annotation, model adaptation, and complement evaluation.

2. The proposed two-phase CAMP framework is well motivated and methodologically uncomplicated, yet demonstrates clear effectiveness as supported by the corresponding ablation studies.

**Weaknesses:**

1. The paper does not summarize the role of the LLMs for this work in a separate section, which contradicts the ICLR 2026 submission policy.

2. The AVA-Captions dataset is entirely generated by a VLM. However, an existing public dataset, RefAVA [1], already provides subject descriptions based on the AVA dataset similar to the AVA-Captions. RefAVA is fully human-annotated and cross-checked, excluding the VLM hallucination risks. Moreover, generating activity descriptions from the original AVA action labels appears somewhat redundant and contributes limited novelty.

3. The evaluation focuses solely on the VQA setup. It remains unclear how the results of the proposed method would compare against combining state-of-the-art atomic video human action recognition models with an additional LLM-based description generation step from predicted activity labels. Furthermore, the evaluation for subject description lacks comparison with lightweight image captioning baselines.

4. There is no ablation study on different backbones. The CAMP framework is only demonstrated on the InternVL3, and the paper does not examine its performance when the proposed masking strategy is extended or adapted to other architectures. This omission limits the claimed generality of CAMP on the VLM.

5. The presentation quality requires improvement. In Fig. 2, the 'green action labels' is confusing; there are typos such as “respentively” (lines 464 & 465); the UNQ in Tab. 6 should be CMP, etc. While the equations in Section 4 are central to the method, several notational choices are ambiguous or unprofessional (e.g., the use of 'dont care', and overlapping conditions like $i \leq j$ and $i \leq T_V + T_Q$).

6. The discussion of failure cases in the VLM-based pipeline is insufficient. For instance, the paper does not analyze errors in the VLM-generated activity descriptions during dataset preparation or inconsistencies in evaluation leveraging different VLMs. As an example, Nova Pro v1 reports higher subject description accuracy for the HC-STVG dataset than for the AVA-Captions dataset on the LLAVA-OV, whereas Qwen2.5-VL produces the opposite outcome under the same setting.

[1] *Referring Atomic Video Action Recognition*, in ECCV 2024.

**Questions:**

1. How was the threshold of 0.75 selected for the ReID similarity score used in person re-identification?

2. What protocol or agreement was used during the manual validation to ensure the quality and correctness of the VLM-generated AVA-Captions?

3. Can the authors also perform the ablation of the two CAMP phases on the AVA-Captions dataset, and provide additional metrics beyond SBert similarity for the bounding-box relaxation ablation to ensure a more unbiased assessment?

4. Should the correct causal condition be $j \leq i$ rather than $i \leq j$, given that $Att$ is denoted as the the matrix for attention mask?

Please respond to both the weaknesses and the questions in the rebuttal phase.

---

> ### Author Response · Authors · 2025-11-21
> **Weakness 1: Role of LLM**
>
> We sincerely thank the reviewer for their valuable questions and we hope to answer them in the comments and the revised paper. Please let us know if we can clarify anything further and we kindly request the reviewer to consider raising their score if our responses satisfactorily address their queries.
>
> LLMs were not used to think of the idea for the paper. For our data annotation and evaluation, we used foundation models and they are duly referenced in the paper. In the revised version, we added a paragraph summarizing the role of LLMs in lines 519-521.

---

> ### Author Response · Authors · 2025-11-21
> **Weakness 2: RefAVA dataset**
>
> As mentioned by the reviewer, RefAVA is an important related work and we added it to the Related work section in the revised version. While RefAVA has no hallucinations, we believe that our proposed dataset also holds complementary value. By extending to open-world activity captions, we are able to capture person-person and person-object interactions that are not covered by RefAVA. For example, consider Figure 6 in the paper. As an example, in the second row, our caption reads "Woman in a blue dress and straw hat: sits on a bicycle, rides through a grassy field, stops to read a book, and continues riding". Across the video, these actions indeed happen. From the original labels or labels in RefAVA, we can never get information about the bicycle, or that she is riding through a grassy field. Hence, our captions add value by adding such details and interactions.

---

> ### Author Response · Authors · 2025-11-21
> **Weakness 3: Comparison with Image Captioning**
>
> We think that using state-of-the-art atomic action detection followed by LLM to patch them up into a coherent sentence would not capture the person-object relationships that are predicted when directly passing the video to the VLM. Similarly, image captioning baselines provide captions for the whole scene but are unable to caption a given subject in a multi-subject setting and provide captions for the whole scene. Following the suggestion of the reviewer, we instead tried employing referring image captioning, where the frame and bounding box around a person are passed to the model, and it outputs a caption describing the person. We used the Describe Anything Model [1] for this purpose. We found that on the AVA dataset, we got an SBert similarity of 0.41 and a METEOR score of 22.95. On the HC-STVG dataset, we got an SBert similarity of 0.35 and a METEOR score of 10.98. However, we don't think this a valid comparison since (i) DAM is not trained specifically on the dataset and (ii) the methods in Table 2 perform multiple tasks (subject and activity description) unlike dedicated captioning models. Directly comparing these fundamentally different types of models may therefore be misleading.
>
> [1] Describe Anything: Detailed Localized Image and Video Captioning - ICCV 2025

---

> ### Author Response · Authors · 2025-11-21
> **Weakness 4: Additional backbone for CAMP**
>
> In the revised version, we added QwenVL2.5-3B baseline and applied CAMP to it. The updated results are in Tables 2,3, 5 and 6. Interestingly, we observe greater relative improvements for the smaller InternVL3-1B backbone compared to the larger Qwen2.5VL-3B. This is expected since larger models can already capture complex functions, making constrained training less impactful, whereas smaller models benefit more from CAMP’s inductive bias. This discussion has been added to the revised version in lines 364-371

---

> ### Author Response · Authors · 2025-11-21
> **Weakness 5: Presentation quality**
>
> We have improved Figures 2 and 6 and updated the caption to read "red action labels" so that our annotations can be clearly distinguished from the original captions. Typos have been corrected. UNQ has been replaced with CMP in Table 6. Equations have been updated. Don't cares have been replaced with 0 and conditions have been corrected in Equations 2 and 3. We would like to apologize for these mistakes and thank the reviewer for helping us improve the presentation quality.

---

> ### Author Response · Authors · 2025-11-21
> **Weakness 6: Failure Cases**
>
> We have added a discussion on limitations in the revised paper in lines 200-208 for VLM-based annotation and in lines 514-518 and Appendix section F for VLM-based evaluation. For annotation, we found out through manual inspection that while not perfect, the VLM is able to satisfactorily generate captions. The major limitation instead is that ReID is not enough to filter out all repeated entities. We believe manual annotation is required for this. For evaluation, we found out that VLM is quite pessimistic for CMP and UNQ metrics and assigns low scores to cases where human inspection would grant higher UNQ and CMP numbers. Thus, these metrics should be mainly used for comparison rather than as an absolute metric. We add this discussion to the aforementioned sections in the revised manuscript.

---

> ### Author Response · Authors · 2025-11-21
> **Question 1: 0.75 threshold selection**
>
> We tested with 0.65, 0.75 and 0.85 as the thresholds on randomly selected samples. This threshold controls the merging of subjects that might describe the same person.
> With 0.65 threhold, we found a significant increase in different people getting merged in annotation. With 0.85, we get a significantly higher number of repetitions. Thus, we found that 0.75 was an acceptable threhold for balancing the amount of repetition and false merging. We added this discussion in the revised paper in lines 200-209

---

> ### Author Response · Authors · 2025-11-21
> **Question 2: Details on manual validation**
>
> We manually validated 1,000 randomly sampled datapoints to assess the quality of captions generated by the VLM. For each video, we examined whether:
> 1. All ground-truth action labels were incorporated into the caption,
> 2. Individuals were described accurately,
> 3. The annotated actions were correctly reflected, and
> 4. The same person was consistently assigned a single subject ID.
>
> Our evaluation revealed that in approximately 95% of cases, the VLM successfully utilized all action
> labels and produced accurate person and activity descriptions. Hence, we concluded that the dataset can be used without going through all the examples, since the biases introduced by the VLM are minor and would not significantly affect model development. We add this discussion in lines 193-209 in the revised paper.

---

> ### Author Response · Authors · 2025-11-21
> **Question 3: More complete ablation of the phases of CAMP**
>
> We updated Table 8 to also include ablations with the AVA dataset and added all the classical metrics. From this, we see clear improvements in all metrics using both phases.

---

> ### Author Response · Authors · 2025-11-21
> **Question 4: Correction in the equation**
>
> The correct condition should be $j \leq i$ as pointed out by the reviewer. We have corrected it in the revised version. We would like to apologize for this typo and would like to assure you that it was limited to the paper and was not reflected in the code.

---

### Official Review · Reviewer_bEqi · 2025-10-31

**Soundness:** 3
**Presentation:** 2
**Contribution:** 3
**Rating:** 4
**Confidence:** 4

**Summary:**

This paper introduces the task of Grounded Human-Attributed Description and Activity Recognition and presents a new dataset, AVA-Caption, derived from AVA-Actions. The proposed CAMP method constrains attention to enhance person-centered grounding for better description and activity recognition. A VLM-based evaluation metric is also introduced to assess captions directly against video content. Despite relying on the VLM’s comprehension ability, the work provides a promising direction for human-attributed video understanding.

**Strengths:**

1. The paper defines a new task, proposes a new dataset, and designs corresponding methods and evaluation protocols, forming a complete and coherent framework.
2. The problem identification is accurate. for example, recognizing that existing datasets lack detailed human attribute descriptions and that current evaluation metrics fail to measure prediction quality precisely.

**Weaknesses:**

1. The motivation is somewhat broad. The paper’s main contribution lies in proposing AVA-Caption, a dataset built upon AVA-Action that includes rich human attributes and action descriptions. However, the paper lacks experimental analysis showing the advantages, value, or necessity of this dataset compared to the original one.

2. The writing is not entirely fluent. For instance, “We observed that when we trained a VLM on our dataset, it was often confusing actions done by one person with the other.” is awkward. Some tables also have formatting issues (e.g., Table 1 and Table 7).

3. In terms of methodology, how are the proportions between Phase 1 and Phase 2 training determined? Overemphasis on Phase 1 may make the model focus too much on local details and ignore global context, while overemphasis on Phase 2 could cause PID confusion, potentially affecting overall performance. Moreover, the designed evaluation metric is not convincing: using a large model such as Qwen2.5-VL 32B for absolute scoring introduces biases, hallucinations, and lacks interpretability. Relative or local ranking might be more reliable. Also, the rationale behind evaluating along CRT, CMT, and UNQ dimensions needs clearer justification.

4. In the experiments, Table 2 and Table 3 compare only two baselines — LLAVA-OV 0.5B and InternVL3-1B — which is insufficient for a convincing comparison. As I understand, CAMP refers to InternVL3-1B trained with both Phase 1 and Phase 2, so does the compared InternVL3-1B correspond to a version trained only on Phase 2?

**Questions:**

See the weaknesses

---

> ### Author Response · Authors · 2025-11-21
> **Q1: Ava-Captions vs AVA-Actions:**
>
> We sincerely thank the reviewer for their valuable questions and we hope to answer them in the comments and the revised paper. Please let us know if we can clarify anything further and we kindly request the reviewer to consider raising their score if our responses satisfactorily address their queries.
>
> We want to clarify the following statement by the reviewer: " The paper’s main contribution lies in proposing AVA-Caption, a dataset built upon AVA-Action that includes rich human attributes and action descriptions." Please note that the original Ava-Actions dataset does not have any human attributes, nor does it have rich action descriptions. This dataset only has action labels, as mentioned in section 2. In Ava-Captions, which is the proposed dataset in this work, we generate rich human attributes and action descriptions, both of which are not present in the original dataset. In addition, in the revised manuscript, we present more discussion about the proposed dataset in lines 186-208. Many applications in home assistants and autonomous surveillance systems require the subject and action description proposed in this paper, which are not enabled by using only Ava-Actions, thereby showing the value of the proposed dataset. In the revised manuscript, we reference existing works that can attest to the value of the GHADAR task and, by extension, our dataset, which enables the task (lines 35-48, 135-149).

---

> > ### Author Response · Authors · 2025-11-21
> > **Q3 B: Limitations of VLM**
> >
> > We agree that VLMs for evaluations can insert hallucinations and bias. However, we would like to clarify that the goal of VLM-based metrics is not to replace other classical metrics, but to complement them in comparing one model with respect to the other, rather than absolute evaluation of a single model. In the revised manuscript, we include a discussion on the limitations of using VLM-metrics in lines 515-517 and Appendix F (lines 833-843)

---

> ### Author Response · Authors · 2025-11-21
> **Q2 Paper Writing Improvements:**
>
> We have revised the manuscript to remove confusing phrases. For example, the sentence noted by the reviewer has been changed to: "We observed that when training a VLM on our dataset, the model frequently confused actions performed by different individuals (for example, action done by person B predicted for person A)" in Section 4.4. We also corrected the table formatting in Tables 1 and 7

---

> ### Author Response · Authors · 2025-11-21
> **Q3 A: Proportion between Phase 1 and Phase 2:**
>
> We sample a portion of the training data for validation and continue to train Phase 1 until the minima of the validation set. Then, we remove constraints in Phase 2, reduce the learning rate, and continue to train until the minima is reached on the validation set in Phase 2.

---

> ### Author Response · Authors · 2025-11-21
> **Q3 C: Rationale behind CRT, CMP and UNQ:**
>
> CRT measures the factual alignment of the prediction with the video content. CMP measures whether all relevant actions are captured by the prediction. UNQ measures ambiguity in the predicted caption. We added this to lines 460-467 in the revised manuscript. These three dimensions address shortcomings of classical metrics, which fail to capture the semantic validity of the predicted caption when they are lexically different than the GT.

---

> ### Author Response · Authors · 2025-11-21
> **Q4: Added baseline:**
>
> Yes, the compared InternVL is the same as Phase 2, but with a higher learning rate. We added Qwen2.5VL-3B and CAMP + Qwen2.5VL-3B to the comparison tables in the revised version. More specifically, we updated lines 349-371 and Tables 2, 3, 5 and 6. Interestingly, we observe greater relative improvements for the smaller InternVL3-1B backbone compared to the larger Qwen2.5VL-3B. This is expected since larger models can already capture complex functions, making constrained training less impactful, whereas smaller models benefit more from CAMP’s inductive bias. This discussion has been added to the revised version in lines 364-371

---

> ### Comment · Reviewer_bEqi · 2025-11-26
>
> Thank you for the effort. Some of your responses addressed my earlier concerns. However, I still remain uncertain about the contribution of this new task. From my perspective, it mainly asks for detailed descriptions of different people in a video. So, how is this essentially different from existing fine-grained video descriptions? Current datasets (e.g., [1]) can already provide very detailed annotations, and your task seems to add only bounding boxes. Moreover, current VLMs already perform well on this task, so the core challenge remains unclear to me. Therefore, I keep my score.
>
> [1] Chen H, Wang X, Chen H, et al. Verified: A video corpus moment retrieval benchmark for fine-grained video understanding[J]. Advances in Neural Information Processing Systems, 2024, 37: 40393-40406.

---

> ### Author Response · Authors · 2025-11-26
> **Difference with respect to fine-grained captioning**
>
> 1) Existing fine-grained video description datasets (including [1]) primarily focus on generating detailed captions for entire scenes instead of per-person descriptions. As seen in all figures in the referenced paper, there is a single subject in the video, which makes it look like it is similar to our task. However, the difference can be seen in multi-subject setups. For example, in Figure 4, row 1 in the referenced paper [1], there are two humans and a dog, but the fine-grained video description only captures the actions of the dog. This shift from global scene-level annotation to subject-specific grounded annotation is crucial for applications such as surveillance, healthcare monitoring, and human–robot interaction, where understanding each person’s behavior matters.
>
> 2) Fine-grained video description can contain a lot of information that may not be required for the application. For example, information about the background, the room the subjects are standing in, and so on. This requires additional postprocessing to extract per-subject activity. Moreover, if there are many people in the video, a fine-grained description usually generates a summarized group activity description, or does not capture the activities of all the subjects. Our grounded approach, by leveraging bounding box prompts, directly conditions the model on the subject of interest, producing concise and targeted descriptions. This design gives users control over which entity to describe, avoiding the pitfalls of group-level or overly verbose captions. Thus, the bounding box is an additional user input that is not available in [1], and the model learns to leverage it to produce concise and precise descriptions.
>
> 3) The distinction between fine-grained description and grounded description parallels the difference between “segment everything” and “prompted segmentation” as introduced by Segment Anything Model (SAM) [2]. While one can theoretically segment all objects in an image, extracting the mask of a user-specified object from that set is infeasible in practice. Prompted segmentation is recognized as a distinct task because it leverages user input to guide the model. Similarly, our task is not simply “describing all subjects”; it is prompt-driven grounded description, where bounding boxes act as explicit prompts to control the caption generation process. This makes it fundamentally different from existing fine-grained captioning
>
> 4) Our contribution through this work is not only a dataset. We propose:
>
> (i) A method that integrates bounding box prompts into the description pipeline, improving subject-specific fidelity.
>
> (ii) A critical analysis of existing evaluation metrics, highlighting their limitations in multi-person grounded settings, which other reviewers have also noted as a valid and important observation.
>
> (iii) A complementary set of metrics tailored to this task
>
> Together, these contributions establish a coherent framework: a new task definition, a dataset enabling it, a method leveraging bounding box prompts, and an evaluation schema that better reflects semantic correctness in multi-person video understanding. We request that the reviewer consider all the contributions for the final review.
>
> [2] Segment Anything ICCV 2023

---

### Official Review · Reviewer_C74a · 2025-10-31

**Soundness:** 2
**Presentation:** 2
**Contribution:** 2
**Rating:** 4
**Confidence:** 3

**Summary:**

The paper introduces a new task, grounded human-attributed description and activity recognition, accompanied by a dataset (re-annotated AVA), a novel attention mask constraint (CAMP) to better align where a Transformer looks at (local vs global), and a VLM-based evaluation metric.

**Strengths:**

In general, the paper is easy to follow. I appreciate that the authors identified problems with existing evaluation methods without blindly trusting existing metrics.

**Weaknesses:**

Why did the authors not share video examples of their new dataset? This would have been very helpful in better understanding the quality. I have major concerns with regard to the text fidelity: the authors utilize an off-the-shelf method to describe the videos, but this runs the risk of the model incorporating text hallucinations (this is a bit of chicken-egg problem, where, in order to train a model, we have to utilize another model). For example, in Figure 2, on the left side, there is a visible cut in the sequence: how is this handled for the video? This is unclear from the description. I find Figure 4 even more concerning as the GT annotation texts seem to barely match the video frame description. Could the authors comment on how they ensured the quality of the data? From the samples they provide the quality seems to be a major bottleneck.

I wonder if this is also one of the reasons why the text-based evaluation metrics are show poor performance: if the gt text is inaccurate, the model predictions might be ok but they will be compared to “bad” gt. In general, I disagree with the authors claim that the eval issues stem from “lexically different but semantically correct” captions - from the shown samples it seems more like that some of the gt captions are semantically incorrect.

In general,  I feel that the claim that existing methods have flaws is valid, but the authors should show this in a meaningful and reproducible way - showing just two samples (with poor gt) is not sufficient.

I suggest that the authors utilize “Equation” to showcase their full-line equations to make discussing them easier.

Could the authors describe what the “dont care” means in their Attention matrices? This notation is a bit unusual.

**Questions:**

L209 “None” - should this be cursive or is this a typo?

**Details Of Ethics Concerns:**

The proposed dataset heavily builds on an existing dataset (AVA) - I would like the authors to clarify if they will require the citation of the original dataset when citing their dataset.

---

> ### Author Response · Authors · 2025-11-21
> **Q1: Regarding more video examples:**
>
> We sincerely thank the reviewer for their valuable questions and we hope to answer them in the comments and the revised paper. Please let us know if we can clarify anything further and we kindly request the reviewer to consider raising their score if our responses satisfactorily address their queries.
>
> We have provided some video examples in Figure 2 and 6 in the paper. Figure 2 is in the main paper and shows 2 videos while Figure 6 is in the appendix, and has 8 examples. We have edited the figures in the revised version so that our annotations are clearly distinguished from the original labels for a given video.

---

> > ### Comment · Reviewer_C74a · 2025-11-25
> >
> > Thanks a lot for the rebuttal! I would like to see the samples "in motion" in a real video (i.e. as .mp4) - to see how well the new description fit the motion - this would make it easier to see if how strong the model-generated data is "hallucinating".

---

> > > ### Author Response · Authors · 2025-11-25
> > > **Anonymous link to videos**
> > >
> > > The videos are from public datasets HCSTVG and AVA. For the purpose of review, we have downloaded the relevant videos which we used in the figures and uploaded them here: https://limewire.com/d/dJvnl#DnhALMNuWt
> > >
> > > Please keep in mind that this link will expire in one week and is anonymized using file.io. The filenames in the zip folder at this link are according to the figures in the paper for convenience

---

> ### Author Response · Authors · 2025-11-21
> **Q2: Text Fidelity and quality of the data:**
>
> The reviewer raises a valid concern regarding using foundation models for annotation. While not perfect, we believe the dataset does solve the purpose of training models. To ensure that the risk of hallucinations is minimal, we use a large-scale VLM (Nova-Pro) and prompt it to generate captions based on the action labels present in the original AVA-Actions dataset, which are manually annotated. We performed manual inspection of around 1000 samples and found that the generated captions consistently captured all action labels while accurately describing person–object interactions and person descriptions. We have updated lines 186–208 in the main paper to provide additional details on this process.

---

> ### Author Response · Authors · 2025-11-21
> **Q3: Breaks in the video:**
>
> The AVA-Actions dataset consists of movie clips where the camera frequently switches between subjects. These breaks are inherent to the dataset and are therefore included during training. While we do not introduce special mechanisms to handle breaks, the model implicitly learns to maintain the temporal associations despite the discontinuities due to their prevalence in the training data. One issue due to the breaks that we did recognize is that the original AVA-Actions gave different PIDs to the same person when the camera pans away from a person and returns. To reduce this, we introduced ReID-based filtering as described in lines 186-208 of the main paper.

---

> ### Author Response · Authors · 2025-11-21
> **Q4: Inaccurate Ground Truth (GT) text:**
>
> In the original submission, we presented single-frame images, which led to confusion regarding ground-truth accuracy. In the revised version, we replaced these with multi-frame images that better capture the scene context. For example, Figure 4 now clearly demonstrates that the ground-truth descriptions are correct

---

> ### Author Response · Authors · 2025-11-21
> **Q5: More examples of differing, but equally valid GTs and predictions:**
>
> Similar to the previous concern regarding incorrect GTs, we have addressed it by editing the single-frame images with more frames from the same video in the revised version. More specifically, please consider Figure 5, which clearly shows the subject of interest and their actions in the video. It can be observed that the GT and predicted caption are both valid, but lexically different, leading to a lower score. We also added Figures 7 and 8 in the appendix in the revised paper, which highlight this through more examples.

---

> ### Author Response · Authors · 2025-11-21
> **Q6: Paper quality improvements:**
>
> We use \begin{equation} in the revised version so that the equations are numbered. We initially used "don't care" to specify that the attention mechanism does not use these tokens for the computation of the loss, and hence, we are not concerned with their values in the matrix. However, this was a valid cause of confusion, and hence, we replaced "don't care" with 0 in the revised version. The "None" in italics in line 238 in the revised version denotes the Pythonic None.

---

> ### Author Response · Authors · 2025-11-21
> **Q7: Ethics Review:**
>
> We extend the AVA-Actions dataset. Hence, we will require future researchers to cite the original dataset if they use our dataset. We have added this to the ethics statement in the revised version and will also mention this when releasing data and code.

---

### Author Response · Authors · 2025-12-02
**Link to Videos for the paper figures**

Reviewer 1 had asked for videos to verify if the generated captions were happening in the videos. While we added multiple frames from the videos in the paper figures, if the videos still need to be accessed, here is the anonymous video link that will expire in 7 days: https://limewire.com/d/JYhAy#I6zd6ALzUJ

Please note that the previous link will expire in 1 day. Kindly use the above link to access videos

---

### Meta-Review · Area_Chair_n7JU · 2026-01-06

**Summary:**

The submission introduces a new task that generalizes spatiotemporal action localization, namely grounding human attributes and activities from videos. The authors introduced a new method "CAMP", a two-stage training procedure which first constains the attention mask to be "person-centric", then relaxes it to be holistic. The evaluation was performed on two benchmarks, HC-STVG, and AVA-Captions, a new benchmark introduced by the authors. AVA-Captions is derived from the AVA dataset originally introduced for spatiotemporal action localization, but now augmented with "attributes", in the form of VLM-generated captions. The authors also introduce an automatic evaluation protocol, again relying on VLMs. The submission initially received three ratings of "4", and the authors provided rebuttals to the concerns. The AC reads the submission, all reviews, and the responses, and believes that the rebuttals do not address the main concerns from the reviewers. The AC thus recommends rejection of the submission.

**Reviewer Concerns:**

- There is a general concern about the risk of VLM hallucinations due to the nature on how the annotations are collected (by VLMs). Reviewer C74a requested video examples to demonstrate the quality, reviewer HH6R mentioned that an existing benchmark RefAVA serves similar purpose but was manually annotated. Reviewer bEqi further questioned the contributions of the introduced task compared to existing work. The AC reads the responses and believes these issues are fundamental and not fully addressed.
- Reviewer bEqi asked about the portions of phase 1 and phase 2 training, which the authors claimed was decided by a validation set. Additionally bEqi asked about the limitations of VLM-based evaluation, which the authors acknowledged (and the AC agrees that this is another fundamental limitation of the adopted paradigm).
- Reviewer HH6R requested additional baseline (e.g., detection then caption) and ablation studies, which were provided by the authors.

**Reviewer Scores:**

Based on the AC's read, there are remaining, fundamental concerns raised by all reviewers that were not fully addressed by the authors, including the potential risks in relying on LLM/VLMs for annotation and evaluation, the merit of the introduced task and data compared to published work, and the clarity of the presentation. Therefore the AC believes that it was unlikely that the reviewers would raise their scores from "borderline reject" to higher ratings.

---

### Decision · Program_Chairs · 2026-01-26

Reject